# Impacts of Bacteriostatic and Bactericidal Antibiotics on the Mitochondria of the Age-Related Macular Degeneration Cybrid Cell Lines

**DOI:** 10.3390/biom12050675

**Published:** 2022-05-07

**Authors:** Nasim Salimiaghdam, Lata Singh, Mithalesh K. Singh, Marilyn Chwa, Shari R. Atilano, Zahra Mohtashami, Anthony B. Nesburn, Baruch D. Kuppermann, Stephanie Y. Lu, M. Cristina Kenney

**Affiliations:** 1Department of Ophthalmology, Gavin Herbert Eye Institute, University of California Irvine, Irvine, CA 92697, USA; salimiag@hs.uci.edu (N.S.); drlata.singh@aiims.edu (L.S.); mithales@hs.uci.edu (M.K.S.); mchwa@hs.uci.edu (M.C.); satilano@hs.uci.edu (S.R.A.); zmohtash@hs.uci.edu (Z.M.); anesburn@hs.uci.edu (A.B.N.); bdkupper@hs.uci.edu (B.D.K.); sylu@hs.uci.edu (S.Y.L.); 2Cedars-Sinai Medical Center, Los Angeles, CA 90048, USA; 3Department of Pathology and Laboratory Medicine, University of California Irvine, Irvine, CA 92697, USA

**Keywords:** AMD cybrids, mtDNA haplogroups, antibiotics, bacteriostatic, bactericidal

## Abstract

We assessed the potential negative effects of bacteriostatic and bactericidal antibiotics on the AMD cybrid cell lines (K, U and J haplogroups). AMD cybrid cells were created and cultured in 96-well plates and treated with tetracycline (TETRA) and ciprofloxacin (CPFX) for 24 h. Reactive oxygen species (ROS) levels, mitochondrial membrane potential (ΔψM), cellular metabolism and ratio of apoptotic cells were measured using H2DCFDA, JC1, MTT and flow cytometry assays, respectively. Expression of genes of antioxidant enzymes, and pro-inflammatory and pro-apoptotic pathways were evaluated by quantitative real-time PCR (qRT-PCR). Higher ROS levels were found in U haplogroup cybrids when treated with CPFX 60 µg/mL concentrations, lower ΔψM of all haplogroups by CPFX 120 µg/mL, diminished cellular metabolism in all cybrids with CPFX 120 µg/mL, and higher ratio of dead cells in K and J cybrids. CPFX 120 µg/mL induced overexpression of *IL-33*, *CASP-3* and *CASP-9* in all cybrids, upregulation of *TGF-β1* and *SOD2* in U and J cybrids, respectively, along with decreased expression of *IL-6* in J cybrids. TETRA 120 µg/mL induced decreased ROS levels in U and J cybrids, increased cellular metabolism of treated U cybrids, higher ratio of dead cells in K and J cybrids and declined ΔψM via all TETRA concentrations in all haplogroups. TETRA 120 µg/mL caused upregulation of *IL-6* and *CASP-3* genes in all cybrids, higher *CASP-7* gene expression in K and U cybrids and downregulation of the *SOD3* gene in K and U cybrids. Clinically relevant dosages of ciprofloxacin and tetracycline have potential adverse impacts on AMD cybrids possessing K, J and U mtDNA haplogroups in vitro.

## 1. Introduction

The introduction of antibiotics in the 20th century is considered an astounding breakthrough in medicine [1]. Antibiotics cure a wide range of infectious diseases, and they facilitate the success of modern medical procedures such as organ transplantations, surgeries and cancer therapy regarding prevention of opportunistic infections [2]. However, over time, the growing pattern of antimicrobial resistance (AMR) and some remarkable adverse effects resulting from administration or misusage of antibiotics have attracted attention [3]. Fluoroquinolones are antibiotics frequently used in medical and surgical fields, such as ophthalmology. These bactericidal agents (e.g., norfloxacin, lomefloxacin, ciprofloxacin and ofloxacin) have broad-spectrum antibacterial activity, and can be administered via topical, intravitreal and systemic routes [4]. Their principal mechanism of action is inhibiting DNA topoisomerases and preventing bacterial DNA replication by impending mitochondrial DNA replication, transcription and lowering mtDNA copy number [5]. Although fluoroquinolones are well-tolerated agents overall, they have induced some serious side effects in some individuals, such as tendinopathy, cardiac arrythmia and optic neuropathy [6,7,8]. Fife and colleagues conducted a case–control study on ophthalmologic patients exposed to fluoroquinolones, who were observed for an average of 27 months. An association between exposure with fluoroquinolones and risk of retinal detachment was demonstrated [9].

Tetracyclines (e.g., tetracycline, demeclocycline, minocycline and doxycycline) are well-established bacteriostatic agents affecting a wide array of Gram-positive and Gram-negative bacteria [10]. Tetracyclines have been effective in the prevention and treatment of allergic, inflammatory, immune and infectious diseases, such as chronic blepharitis, scleroderma and pyoderma gangrenosum [11,12,13]. Tetracycline derivatives have a high affinity to bacterial ribosome 30S subunit. Therefore, their mechanism of action is blocking the bacterial protein synthesis process by preventing the association between aminoacyl-tRNA and bacterial ribosome 30S subunits [14]. Considering the negative impacts of tetracyclines on matrix metalloproteinases (MMPs), mast cells and production of inflammatory cytokines, studies have demonstrated the possible therapeutic role of these agents against COVID-19 [15,16,17]. In terms of adverse effects of tetracyclines, studies have revealed their potential side effects such as phototoxic dermatitis, gastrointestinal disturbance and discoloration of teeth and oral cavity [18,19]. Wallace et al. reported the reduction in intraocular pressure of cats resulting from intraocular injection of tetracycline derivatives, particularly demeclocycline [20].

Mitochondria are intracellular organelles which are the main source of energy metabolism in eukaryote cells via ATP production [21]. Moreover, mitochondria are involved in numerous essential intracellular reactions such as ROS production, apoptosis, synthesis of lipids and steroids, breaking down of sugars and fatty acid oxidation [22]. The hypothesis of an endosymbiotic origin of mitochondria occurring billion years ago has been discussed in previous studies [23]. Later studies indicated remarkable similarities between bacteria and mitochondria with respect to their sizes, circular DNA, ribosomal subunits and electron transport chain [24,25]. Previous in vitro studies showed the detrimental anti-mitochondrial outcomes of antibiotics. Kalghatgi and colleagues conducted a study using bactericidal and bacteriostatic antibiotics on the peripheral blood cells of mice. Authors treated the mice with kanamycin (25 µg/mL), ampicillin (20 µg/mL) and ciprofloxacin (10 µg/mL) along with tetracycline (10 µg/mL). They found that cells treated with bactericidal agents showed disrupted mitochondrial electron transport chain leading to higher ROS production [26]. Patients suffering from age-related macular degeneration (AMD) showed a higher level of retinal mitochondrial DNA (mtDNA) abnormalities such as deletions [27]. Therefore, the harmful effects of antibiotics on damaged mitochondria might be more crucial in these individuals.

In our previous study conducted on ARPE-19 (human retinal pigment epithelial) cell lines, we found significantly decreased levels of ΔψM and cellular metabolism after TETRA and CPFX treatments. Furthermore, overexpression of apoptotic (*CASP-7*, *CASP-9* and *BAX*) and inflammatory (*IL-6*) pathway genes was induced by exposure to CPFX [28]. Moreover, our recent study performed on MIO-M1 (human retinal Müller) cells treated with CPFX and TETRA showed CPFX induced lower levels of cellular metabolism, ΔψM and ROS along with overexpression of *CASP-3* and *CASP-9* [29]. 

Age-related macular degeneration (AMD) is a retinal degenerative disease, affected by various genetic and environmental risk factors which cause progressive vision loss in the elderly population of developed countries [30,31]. AMD cybrids are cell lines that have identical nuclei but mitochondria from different individuals. Analyses of the mtDNA showed that AMD is associated with the J, T and U haplogroups but individuals with the H haplogroup are at low risk for developing AMD [32]. Mitochondrial DNA variants of respiratory complex I that uniquely characterize haplogroup T2 are associated with increased risk of age-related macular degeneration [33]. Kenney at al. demonstrated that mtDNA haplogroups have key roles in numerous molecular pathways associated with AMD progression such as energy production, cell growth and signaling [34].

In this study, our aim is to assess the potential damaging effects of CPFX and TETRA on the cellular health and mitochondria of AMD cybrid cell lines.

## 2. Materials and Methods

### 2.1. Ethic Statement

This study involving human subjects was performed according to stated principals in Declaration of Helsinki. After obtaining the informed written consent, the Institutional Review Board of the University of California, Irvine approved the research (UCI IRB #2003-3131).

### 2.2. Cybrids Creation

The process of creation of trans-mitochondrial cybrids is described in previous studies [33]. After collecting 10 mL of peripheral blood using tubes with sodium citrate, DNA extraction kits (Puregene, Qiagen, Valencia, CA, USA) and the Nanodrop 1000 (Thermo Scientific, Wilmington, DE, USA) were used for DNA isolation and quantification, respectively. Then, a series of centrifuge steps were performed for platelet isolation followed by using Tris buffer saline (TBS) for platelet suspension. ARPE-19 cell lines, purchased from ATCC (Manassa, VA, USA), demonstrated similar functional and structural characteristics to in vivo RPE cells [35]. The ARPE-19 were exposed to low-dose ethidium bromide and serially passaged until they were deficient in mtDNA (Rho0) [36]. As described in previous studies, cybrids were created by fusion of Rho0 ARPE-19 cells with platelets using polyethylene glycol [32]. 

### 2.3. Cybrids Culture Conditions

For culturing the cybrids, DMEM-F12 serum media contained 10% dialyzed fetal bovine serum, 17.5 mM glucose, 2.5 µg/mL fungizone, 100 µg/mL streptomycin, 50 µg/mL gentamycin, 100 unit/mL penicillin and 50 µg/mL gentamycin. Experiments examined a total of 8 cybrid cell lines that originated from different individuals with AMD (K cybrids, *n* = 3; U cybrids, *n* = 3 and J cybrids, *n* = 2). For all experiments, only passage 5 cybrid cell lines were used. After being plated, the cultures were not treated until the cybrid cells reached confluency. Table 1 illustrates the epidemiological information of the cybrids used in this study.

### 2.4. Intracellular Level of Reactive Oxygen Species (ROS Assay)

For performing ROS, JC-1 and MTT assays, the cybrid cell lines were placed in 96-well plates (10^4^ cells/well) and incubated in 37 °C for 24 h. Then, cells were treated with antibiotics and cultured for an additional 48 h. ROS was measured by adding 100 µL/well H2DCFDA solution (2′,7′-dichlorhydrofluorescin diacetates; Catalog# D399, Thermo Fisher Scientific, Waltham, MA, USA). The presence of intracellular ROS induces conversion of the dye to fluorescent molecules. The fluorescent plate reader (SoftMax Pro, version 6.4, Catalog# 94089, Sunnyvale, CA, USA) was used to measure the excitation (492 nm) and emission (520 nm) wavelengths. For the ROS assay, there were four TETRA groups: vehicle-control (Meth), TETRA (30 µg/mL, 60 µg/mL and 120 µg/mL); and four CPFX groups: Vehicle-control (HCl), CPFX (30 µg/mL, 60 µg/mL and 120 µg/mL). Entire experiments were repeated 3 times.

### 2.5. Mitochondria Membrane Potential (ΔψM) (JC-1 Assay)

Cells were cultured in 96-well plates (10^4^cells/well). After adding the JC-1 reagent (5,5′,6,6′-tetrachloro1,1′,3,3′-tetraethyl-benzimidaz olylcarbocyanine iodide; Catalog# 30001, Biotium, CA, USA) to each well, plates were read using the fluorescent plate reader (SoftMax Pro, version 6.4, Catalog# 94089, Sunnyvale, CA, USA). The green (EX 485 nm and EM 535 nm) and red (EX 550 nm and EM 600 nm) emissions were used for measuring the ratio of red to green fluorescent. For the JC-1 experiments, there were four TETRA groups: vehicle-control (Meth), TETRA (30 µg/mL, 60 µg/mL and 120 µg/mL); and four CPFX groups: Vehicle-control (HCl), CPFX (30 µg/mL, 60 µg/mL and 120 µg/mL). Entire experiments were repeated 3 times.

### 2.6. Cellular Metabolism Assay (MTT Assay)

For measuring levels of cellular metabolism, the MTT assay was performed using 96-well plates. An amount of 10 µL MTT assay reagent (3-(4,5-Dimethyltiazol-2-yl)-2,5-dipheniltetrazolium bromide; Catalog# 30006, Biotium, CA, USA) was added to each well containing cultured cybrids (10^4^/well). After 2 h incubation in 37 °C, 100 µL DMSO was added to each well. Then, Biotek Elx808 Absorbance Reader, (Winooski, VT, USA) was used to analyze the plates. For the MTT experiments, there were four TETRA groups (vehicle-control (Meth), TETRA (30 µg/mL, 60 µg/mL and 120 µg/mL)) and four CPFX groups (vehicle-control (HCl), CPFX (30 µg/mL, 60 µg/mL and 120 µg/mL)). Experiments were repeated 3 times.

### 2.7. The Ratio of Live, Apoptotic and Dead Cells (Flow Cytometry)

To measure the ratio of live and apoptotic cells affected by exogenous 120 µg/mL CPFX and TETRA treatments, cells were cultured in 6-well plates (50,000/well) and incubated for 24 h. After 48 h treatment period with antibiotics, 50 µL propidium iodide (PI, representing dead cells) and YO-PRO-1 (an early marker for apoptosis) stains were added to seeded cells. Cells were incubated for 30 min on ice followed by flow cytometry analyses (Flow Cytometer, Novocyte 3000; Catalog# 4355488, San Diego, CA, USA). The flow cytometry values were measured using 488 nm excitation with green fluorescence emission for YO-PRO^®^-1 dye (i.e., 530/30 bandpass) and red fluorescence emission for propidium iodide (i.e., 610/20 bandpass). This technique measures the physical and chemical characteristics of the cells such as the proportion of live, apoptotic and dead cells in untreated, vehicle-control and antibiotic-treated (CPFX and TETRA) groups. To accomplish standard compensation, single-colored stained cells were used. To assure a statistically significant determination of a sample volume, 10,000 bead events were collected per each sample.

### 2.8. Isolation of RNA and cDNA Amplification 

Cultured cybrids in 6-well plates were treated with 120 µg/mL CPFX and TETRA. After 48 h incubation period, RNA was isolated from the cell lysis using the RNeasy Mini-Extraction kit (Puregene, Qiagen, Valencia, CA, USA). Nano Drop 1000 (Thermo Scientific, Wilmington, DE, USA) was used for RNA quantification. Then, each isolated RNA was reverse transcribed to complementary DNA (cDNA) by using Superscript IV VILO Master Mix with the DNase Enzyme (ThermoFisher, Waltham, MA, USA). 

### 2.9. Quantitative Real Time Polymerase Chain Reaction (qRT-PCR)

Isolation of total RNA was performed from cultured CPFX-exposed (*n* = 3), TETRA-exposed (*n* = 3), associated vehicle-control (HCl and TETRA) (*n* = 3) and untreated (*n* = 3) cells. Table 2 demonstrates all target primers, which are predesigned via Qiagen QuantiTect Primer Assays or KiCqStart SYBR^®®^ Green primers (Sigma–Aldrich, Burlington, MA, USA). PowerUp SYBR Green Master Mix (ThermoFisher) on a ThermoFisher Quant Studio 3 Real-Time PCR System was used for performing qRT-PCR, which evaluated the relative expression level of various genes associated with pro-inflammatory (*IL-6, IL-33* and *TGF-β1*), pro-apoptotic (*CASP-3*, *CASP-7* and *CASP-9*) and antioxidant enzyme (*SOD2*, *SOD3* and *GPX3*) pathways genes. In this study, the housekeeping gene was *HPRT1*, which is an enzyme recycling inosine and guanine in the purine salvage pathway and is considered a stable endogenous control gene when investigating alterations in gene expression. Therefore, as a consistent endogenous control, *HPRT1* primer was used as the reference gene for reaching a standard expression level for all primers [37]. The ^ΔΔ^Ct method was used for analyzing the data, in which ^Δ^Ct = [Ct (threshold value) of the target gene] − [Ct for HPRT1], and ^ΔΔ^Ct = ^Δ^Ct of the treatment condition − ^Δ^Ct of the untreated condition. When comparing the treated conditions to untreated conditions, the fold changes were measured as: fold change = 2^−ΔΔCt^. Triplicate formats of antibiotic treated cells (CPFX and TETRA) compared to untreated and vehicle-control (HCl and Meth) samples were analyzed.

### 2.10. Statistical Analyses

GraphPad Prism (Version 5.0, GraphPad Software, Inc., San Diego, CA, USA) was used for statistical analyses. One-way ANOVA Tukey test was performed when comparing the differences among untreated, vehicle-control (HCl and Meth) and antibiotic-treated (CPFX and TETRA) cells. The *p* value of less than 0.05 was considered statistically significant. Three replicates were allocated for each condition in this study and the associated data analyzed in triplicate.

## 3. Results

### 3.1. ROS Levels

First, we compared the effect of antibiotics on the oxidative stress in the cells of different cybrid haplogroups. In terms of intracellular ROS levels, the different mtDNA haplogroups cybrids demonstrated varying results. After treatment with 60 μg/mL CPFX, U cybrids showed increased ROS levels (*p* < 0.05) (Figure 1b). In contrast, CPFX-treated J cybrids showed decreased ROS levels caused by CPFX 120 µg/mL (*p* < 0.05), (Figure 1c). Similarly, TETRA 120 µg/mL decreased ROS levels (*p* < 0.001) in J cybrids (Figure 1c). Furthermore, neither of the antibiotics affected the ROS levels in treated K haplogroups (Figure 1a). These results indicate that higher dosages of these antibiotics could have impacts on the ROS levels in U and J haplogroups. 

### 3.2. Alterations of Mitochondrial Membrane Potential (ΔψM)

Next, we investigated the impact of antibiotics on the ΔψM of different cybrid haplogroups. The ΔψM levels were decreased by both CPFX and TETRA treatments. The CPFX 120 µg/mL decreased ΔψM in K cybrids (*p* < 0.0001) (Figure 1d). The U cybrids treated with 60 (*p* < 0.05) and 120 µg/mL CPFX (*p* < 0.0001) showed decreased ΔψM, respectively (Figure 1e). Moreover, ΔψM levels were lower in J cybrids treated with CPFX 120 µg/mL (*p* < 0.0001) (Figure 1f). All TETRA treatment concentrations caused significant reduction in ΔψM in all cybrid groups. In K cybrids treated with TETRA 30, 60 and 120 µg/mL (*p* < 0.0001), there was a remarkable reduction in ΔψM (Figure 1d). Furthermore, ΔψM levels were decreased in U cybrids via TETRA 30 µg/mL (*p* < 0.05) along with TETRA 60 and 120 µg/mL (*p* < 0.0001) (Figure 1e). Moreover, J cybrids treated with TETRA 30, 60 and 120 µg/mL (*p* < 0.0001) showed significantly reduced levels of ΔψM (Figure 1f). These results demonstrated that higher treatment concentrations of CPFX and all treatment concentrations of TETRA negatively affected the exposed K, U and J cybrids. 

### 3.3. Changes in Cellular Metabolism (MTT Assay)

We then investigated whether antibiotic treatment has an influence on the cellular metabolism of a cybrid with different haplogroups. In cultured K cybrids, CPFX 30 μg/mL (*p* < 0.01), 60 μg/mL (*p* < 0.0001) and 120 μg/mL (*p* < 0.0001) decreased cellular metabolism (Figure 1g). Exposure to CPFX 120 µg/mL (*p* < 0.01) decreased cellular metabolism of U cybrids (Figure 1h). Interestingly, the U cybrids showed higher cell metabolism when treated with TETRA 120 µg/mL (*p* < 0.05) compared to the vehicle-control-treated cybrid (Figure 1h). When J cybrids were treated with CPFX 30, 60 or 120 μg/mL, there were significantly diminished levels of cellular metabolism (*p* < 0.0001) (Figure 1i). The TETRA 30 μg/mL also lowered the metabolism of exposed cybrids (*p* < 0.0001) (Figure 1i). These results demonstrate that CPFX and TETRA have detrimental effects on the cellular metabolism.

### 3.4. Ratio of Apoptosis and Dead Cells (Flowcytometry)

We investigated whether apoptotic cell death contributes to the negative impact of antibiotics on the different haplogroup cybrids. In K cybrids, treatment with 120 µg/mL of CPFX induced 100% value dead cells (*p* < 0.0001), versus 9.53% dead cells induced by vehicle-control treatment (Figure 2a,d). The TETRA-treated cells showed 18.66% (*p* < 0.05) dead cells versus 15.98% for the vehicle-control treated cells (Figure 2g,j). In U cybrids, there was no significant alteration of the ratio of dead cells with neither CPFX (17.81%) (Figure 2b,e) and TETRA (25.12%) (Figure 2h,k) versus HCl (15.64%) and Meth (23.61%)-exposed cells, respectively. Compared to vehicle-control-exposed cells, the J cybrids treated with CPFX 120 µg/mL showed a substantial increase in relative dead cells (90.52% versus 25.40%) (*p* < 0.05) (Figure 2c,f). The TETRA-treated cells also showed an increase in dead cells (16.36% versus 6.47%) (*p* < 0.05) compared to vehicle-control treated cells (Figure 2i,l). Therefore, CPFX and TETRA significantly increase the proportion of apoptotic and finally dead cells. 

### 3.5. Alterations of Pro-Inflammatory, Pro-Apoptotic and Antioxidant Enzymes Genes

Next, we evaluated the effects of the antibiotics on the different cybrids at the gene level.

#### 3.5.1. K Cybrids

Inflammation genes: Performing qRT-PCR, in all experiments, the value of 1 is assigned to the vehicle-control samples. In terms of the pro-inflammatory gene pathway in K cybrids, the relative gene expression of *IL-33* (*p* < 0.0001) (Figure 3b) increased significantly with CPFX treatment. There was higher expression of *IL-6* gene (*p* < 0.0001) (Figure 3a) with TETRA treatment compared to vehicle-control samples. However, TETRA treatment showed no effect on the *IL-33* (Figure 3b) and *TGF-β1* gene expression levels (Figure 3c) compared to the vehicle-control group. Additionally, there was no alteration of *TGF-β1* gene after CPFX treatment (Figure 3c).

Pro-apoptosis genes: In K cybrids, CPFX stimulated overexpression of *CASP-3* (*p* < 0.0001) (Figure 4a), *CASP-7* (*p* < 0.0001) (Figure 4b) and *CASP-9* (*p* < 0.0001) (Figure 4c), in comparison with vehicle-control samples. Similarly, after TETRA treatment, expression levels of *CASP-3* (Figure 4a) and *CASP-7* (Figure 4b) of K cybrids were upregulated (*p* < 0.01).

Antioxidant genes: The CPFX-treated K cybrids showed no significant impacts on the expression levels of *SOD1* (Figure 5a), *SOD2* (Figure 5b), *SOD3* (Figure 5c) and *GPX3* (Figure 5d) genes. With TETRA treatment, there were lower expression levels of *SOD3* (*p* < 0.01) (Figure 5c) and *GPX3* (*p* < 0.05) (Figure 5d) genes in K cybrids versus vehicle-control-exposed samples. 

#### 3.5.2. U Cybrids

Inflammation genes: The CPFX-treated U cybrids showed upregulation of *IL-33* (*p* < 0.01) (Figure 3e) and *TGF-β1* (*p* < 0.05) (Figure 3f), compared to vehicle-control cells. Although TETRA induced a higher expression level of *IL-6* (*p* < 0.01) (Figure 3d), the expression level of *IL-33* (Figure 3e) and *TGF-β1* (Figure 3f) genes was not affected after TETRA treatment in U cybrids. 

Pro-apoptosis genes: In comparison to vehicle-control cells, treated U cybrids with CPFX showed significantly higher expression of *CASP-3* (*p* < 0.01) (Figure 4d) and *CASP-9* (*p* < 0.05) genes. However, there was no alteration of expression level of *CASP-7* (Figure 4e) in CPFX-treated U cybrids. Furthermore, TETRA-treated U cybrids showed upregulation of *CASP-3* (*p* < 0.01) (Figure 4d) and *CASP-7* (*p* < 0.05) (Figure 4e) genes. The expression of *CASP-9* (Figure 4f) showed no significant alteration after TETRA treatment.

Antioxidant genes: The U cybrids showed higher expression levels of *SOD2* gene (*p* < 0.01) (Figure 5f) via TETRA. However, TETRA caused downregulation of *SOD3* gene in exposed U cybrids (*p* < 0.05) (Figure 5g). CPFX exposure did not affect expression levels of *SOD2* (Figure 5f), *SOD3* (Figure 5g) and *GPX3* (Figure 5h) genes. Additionally, neither treatment altered the expression level of *SOD1* gene (Figure 5e).

#### 3.5.3. J Cybrids

Inflammation genes: The CPFX-treated J cybrids showed upregulation of *IL-33* (*p* < 0.01) (Figure 3h) but lower expression levels of *IL-6* (*p* < 0.0001) (Figure 3g) along with no effects on *TGF-β1* (*p* = 0.013) (Figure 3i) genes. The TETRA-treated J cybrids showed an increased expression level of *IL-6* gene (*p* < 0.0001) (Figure 3g) along with downregulated expression of *TGF-β1* (*p* < 0.01) (Figure 3i) and no alteration of *IL-33* (Figure 3h) genes. 

Pro-apoptosis genes: CPFX and TETRA treatment of J cybrids, increased expression level of *CASP-3* (*p* < 0.05) (Figure 4g) and *CASP-9* (*p* < 0.0001) (Figure 4i) genes, compared to vehicle-control group. Neither antibiotic altered the expression level of *CASP-7* gene in treated J cybrids (Figure 4h).

Antioxidant genes: The CPFX-treated J cybrids showed substantial overexpression of *SOD2* (*p* < 0.05) (Figure 5j) and *SOD3* (*p* < 0.01) (Figure 5k) along with no alteration of *SOD1* (Figure 5i) and *GPX3* (Figure 5l), respectively, compared to vehicle-control samples. The TETRA treatment induced a higher expression level of *SOD2* (*p* < 0.01) (Figure 5j) and *GPX3* (*p* < 0.05) (Figure 5l) genes but no alteration of expression level of *SOD1* (Figure 5i) and *SOD3* genes (Figure 5k). 

These results demonstrate that CPFX and TETRA treatments affect the relative expression levels of various genes associated with inflammatory, apoptotic and antioxidant enzymes. Figure 6 summarizes the expression levels of genes associated with apoptotic, inflammatory and antioxidant enzymes after CPFX and TETRA treatments. 

## 4. Discussion

Fluoroquinolones and tetracyclines have a wide range of antibacterial activities but previous studies have demonstrated possible ocular toxicities as a result of fluoroquinolone administration. In Canada, Etminan et al. conducted a case–control study on 989,591 patients with ophthalmic disease, who were followed between 2000 and 2007. They reported a higher risk of retinal detachment in patients who used oral fluoroquinolones (3.3% of cases) compared to the control group (0.6%) [38]. Another group concluded that the hazard of uveitis-associated systemic illnesses was higher in individuals treated with fluoroquinolones in the outpatient setting [39]. These findings suggest that the mitochondrial sensitivities to these antibiotics may play a role in the side effects seen in some patients. The present study investigates the effects of CPFX and TETRA on cybrid cell lines that possess different mtDNA haplogroup patterns.

## 5. Effects of CPFX on AMD Cybrids

**Mitochondrial membrane potential and ROS levels after CPFX treatment:** Our findings indicate that different cybrid haplogroups respond differently to treatment with CPFX; (i) There is a reduction in ROS levels in J haplogroup after treatment with CPFX 120 µg/mL; (ii) all haplogroup cybrids showed decreased ΔψM levels and lower cell metabolism with CPFX 120 μg/mL treatment; additionally, (iii) all three haplogroups treated with CPFX 120 µg/mL showed significantly elevated ratios of apoptotic and dead cells. 

Other studies have also reported mitochondrial dysfunction and higher ROS levels after antibiotic exposure. Kalghatgi et al. assessed the effects of three bactericidal antibiotics (kanamycin, ciprofloxacin and ampicillin) and one bacteriostatic (tetracycline) on harvested mouse mammary tissues [26]. The bactericidal agents negatively affected mitochondria electron transportation chain (ETC), which led to significantly increased ROS levels and eventually cellular lipid, protein and DNA damage [26]. In another study, application of higher concentrations of fluoroquinolones and tetracycline impaired mitochondrial function in cultured primary human osteoblasts [40]. Similar harmful effects on mitochondria were also shown in animal studies. For example, intraperitoneal injection of fluoroquinolones (alatrofloxacin) caused oxidative stress and mitochondrial injury in a Sod2+/− (heterozygous deficiency in mitochondrial superoxide dismutase, Sod2) mouse model. Compared to wild-type mice, there was a significant reduction in mitochondrial genes and critical enzyme activities in mutant models after injection of 33 mg/kg/day of alatrofloxacin over 28 days. Moreover, the activation of mitochondrial Ca^2+^ dependent mechanisms, such as nitric oxide (NOS) production, was more prominent in alatrofloxacin-treated mutant mice [41]. Multiple studies reported substantial structural and functional alterations of mitochondria in AMD subjects, including disorganized membranes, cristae and matrix along with reduction in mitochondria numbers [32,42,43]. Therefore, this harmful role of bactericidal agents could be more prominent in individuals suffering from an impaired antioxidant defense system along with patients with specific age-related disease with damaged mitochondria such as AMD and Parkinson’s disease. By identifying eukaryotic mitochondrial dysfunction and over-production of ROS linked to fluoroquinolones, therapeutic approaches are progressing regarding alleviating the harmful side consequences of antibiotics administration [44].

**Gene expression studies after CPFX treatment:** Exposure with CPFX 120 µg/mL induced (i) overexpression of *IL-33* in all treated haplogroups; and (ii) upregulation of *CASP-3* and *CASP-9* genes in all exposed haplogroups. Prior literature has shown considerable retinal damage due to increased phototoxicity after fluoroquinolone intake. The phototoxicity of five different fluoroquinolones (ciprofloxacin, enoxacin, DR-3355 (the s-isomer of ofloxacin), lomefloxacin and ofloxacin) was demonstrated in culture mouse 3T3 fibroblast cells exposed to ultraviolet-A (UVA) light [45].

Cellular apoptosis has been described as the mechanism associated with the cytotoxic proprieties of fluoroquinolones. Kohanski et al. compared the ROS levels in Escherichia coli produced via three various categories of bactericidal agents including 250 ng/mL norfloxacin, 5 µg/mL kanamycin and 5 µg/mL ampicillin compared to bacteriostatic ones including tetracycline, erythromycin, spectinomycin and chloramphenicol treatments. It was indicated that the bactericidal groups induced higher deleterious production of hydroxyl radicals contributing oxidative damage and cellular death [46]. To our knowledge, this is the first study to measure the effects of CPFX treatment on expression patterns of genes associated with basic pathways.

### Effects of TETRA on AMD Cybrids

**Mitochondrial membrane potential and ROS levels after TETRA treatment**: Our results demonstrated: (i) significantly diminished ROS levels in U and J haplogroup cybrids via TETRA 120 µg/mL; (ii) declining levels of ΔψM with all TETRA treatment concentrations in all three haplogroups; (iii) diminished cellular metabolism of treated K and J haplogroups cybrids after TETRA 30 µg/mL; and (iv) significant increase in the ratio of dead cells of K and J haplogroups treated with TETRA 120 µg/mL. Similar detrimental effects have been reported by Ding et al. who showed the toxicity of a series of concentrations (0, 45, 60 and 90 mg/L) of two types tetracyclines (doxycycline and chlortetracycline) and four fluoroquinolones (enrofloxacin, norfloxacin, ciprofloxacin and ofloxacin) on wild-type zebrafish. It was suggested that over 7-, 14- and 21-day exposure periods, there was a significant decrease in mitochondria crista along with severe mitochondria swelling in the treated subjects [47]. Another study in agreement with our results investigated the role of doxycycline on the proliferation and metabolism of in vitro breast epithelial lines (MCF12A) [48]. Moreover, the similar risk and mitochondrial-damaging mechanisms are identified as a consequence of topical administration of tetracyclines [49].

**Expression studies after TETRA treatment:** Our findings showed cybrids treated with TETRA 120 µg/mL had (i) up-regulation of *IL-6* and *CASP-3* levels in all haplogroups; but (ii) down-regulation of *IL-33 and SOD3* genes in K and U haplogroup cybrids. Ahler et al. studied the impacts of doxycycline on alteration of proliferation and metabolism of multiple human cell lines in vitro, such as cervical cancer cells (Hela), lung cancer cells (H157) and prostate cancer cells (LNCaP). There was a widespread alteration of expression of genes associated with the oxidative metabolic pathway along with dominancy of the glycolytic metabolism pathway [50]. They examined expression levels of genes associated with gluconeogenesis (fructose-1,6-bisphosphatase (*FBP*) and glucose-6-phosphatase (*G6PC*)) along with regulatory enzymes of the glycolysis pathway (pyruvate kinase (*PK*), hexokinase (*HK*) and phosphofructokinase (*PFK*)). They reported downregulation after TETRA treatment of the *G6PC* and *FBP* genes, along with upregulatory effects on *PK* gene levels [49]. In our study, the principal enzymes related to the antioxidant pathway were affected by treatments. TETRA 120 ug/mL induced downregulation of *SOD2* expression in U and J haplogroups along with lower expression of *GPX3* in exposed K haplogroups. Noteworthily, Jones et al. reported the importance of the disruptive role of tetracyclines in individuals with mitochondrial defects. The fibroblast cells derived from four categories of patients with defective mitochondrial diseases were exposed to doxycycline (10 µg/mL) or tetracycline (77 µg/mL). There was significant disruption of mitochondrial translation resulting from exposure to these antibiotics [50].

## 6. Conclusions

In light of these results, the clinically relevant dosages of bactericidal and bacteriostatic agents showed detrimental effects on treated in vitro AMD cybrid cell lines. The negative association of antibiotics and mitochondria might be important in populations suffering from age-related disorders that are associated with damaged mitochondria, such as AMD, Parkinson’s and Alzheimer’s diseases. In the future, additional in vivo and clinical trials may need to be conducted to improve our understanding and knowledge regarding the negative contributory roles of these antibiotics on mitochondrial dysfunction.

## 7. Strength and Limitations of the Study

The findings of this study might be significant in clinical settings, particularly regarding the approach to treat the patients suffering from specific age-related disease such as AMD, Parkinson’s disease and Alzheimer’s disease. The only limitation of the study is the smaller cohort of individuals with AMD for three different cybrid cell lines.

## Figures and Tables

**Figure 1 biomolecules-12-00675-f001:**
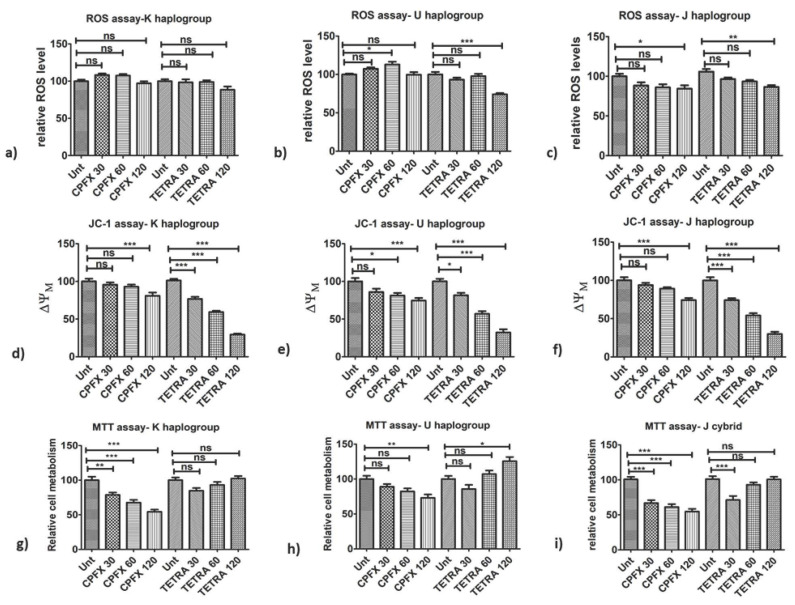
Impacts of CPFX and TETRA on ROS levels, ΔψM and cellular metabolism of K, U and J cybrids via ROS, JC-1 and MTT assays. (**a**) no changes in ROS levels via neither antibiotics in K cybrids; (**b**) CPFX 60 µg/mL increased ROS levels in U cybrids and diminished via TETRA 120 µg/mL; (**c**) ROS levels reduced by CPFX and TETRA 120 µg/mL in J cybrids; (**d**) CPFX 120 µg/mL and all TETRA treatments reduced ΔψM in K cybrids; (**e**) ΔψM diminished via CPFX 60 and 120 µg/mL and all TETRA concentrations in U cybrids; (**f**) CPFX 120 µg/mL along with all TETRA concentrations decreased ΔψM in J cybrids; (**g**) Diminished cellular metabolism with all CPFX concentrations in K cybrids; (**h**) CPFX 120 µg/mL reduced cellular metabolism of U cybrids but higher level with TETRA 120 µg/mL; (**i**) All CPFX treatments and TETRA 30 µg/mL declined cellular metabolism of J cybrids. (* *p* < 0.05, ** *p* < 0.01, *** *p* < 0.0001).

**Figure 2 biomolecules-12-00675-f002:**
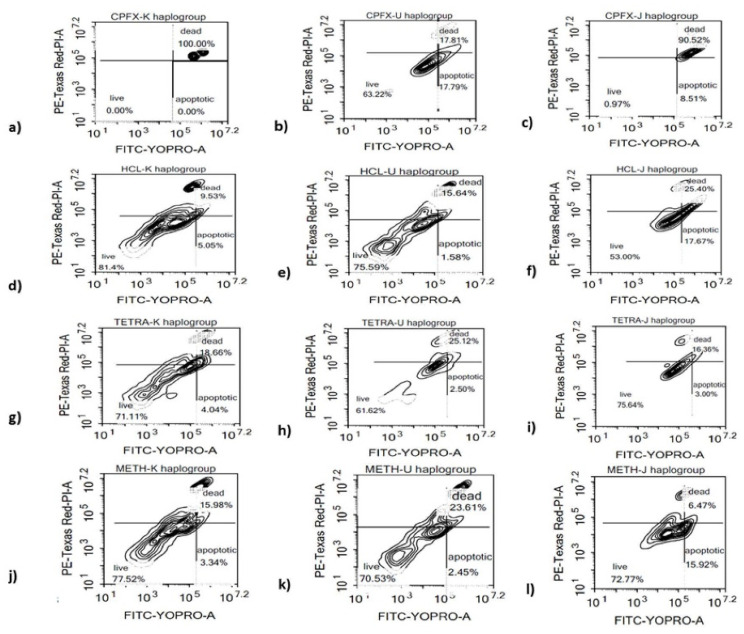
Effects of CPFX and TETRA on the ratio of live, apoptotic and dead cells of K, U and J cybrids. (**a**,**d**,**g**,**j**) higher relative ratio of apoptotic and dead K cybrids via CPFX and TETRA; (**b**,**e**,**h**,**k**) No significant change in ratio of apoptotic and dead cells via either treatment in U cybrids; (**c**,**f**,**i**,**l**) elevated ratio of apoptotic and dead J cybrids vi CPFX and TETRA.

**Figure 3 biomolecules-12-00675-f003:**
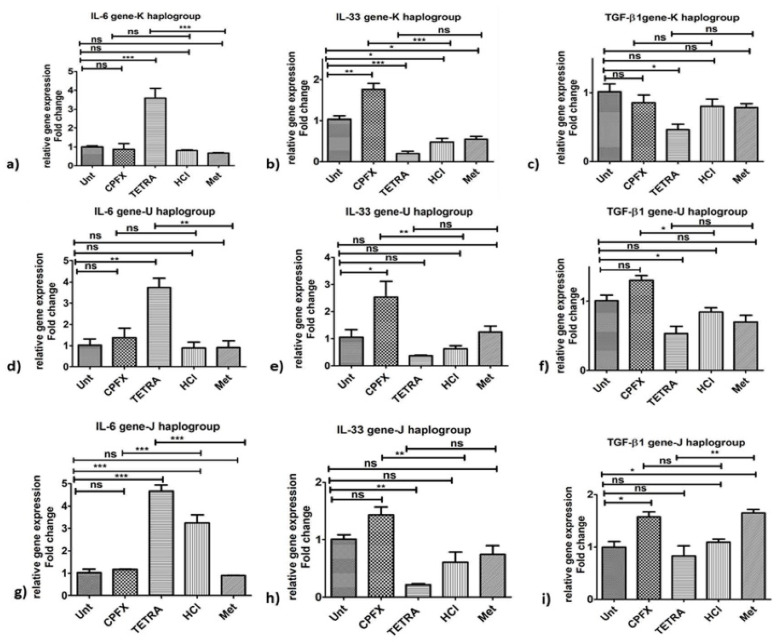
Changes in pro-inflammatory gene expression by CPFX and TETRA in K, U and J cybrids. In K Cybrids, (**a**) overexpression of *IL-6* via TETRA, (**b**) higher *IL-33* expression via CPFX; (**c**) and higher *TGF-β1* expression via CPFX (**c**). In U cybrids, (**d**) higher *IL-6* expression via TETRA, (**e**) higher *IL-33* expression via CPFX and (**f**) upregulation of *TGF-β1* via CPFX. In J Cybrids, (**g**) lower expression of *IL-6* via CPFX, (**h**) upregulation of *IL-33* via CPFX; (**i**) and downregulation of *TGF-β1* via TETRA. (* *p* < 0.05, ** *p* < 0.01, *** *p* < 0.0001).

**Figure 4 biomolecules-12-00675-f004:**
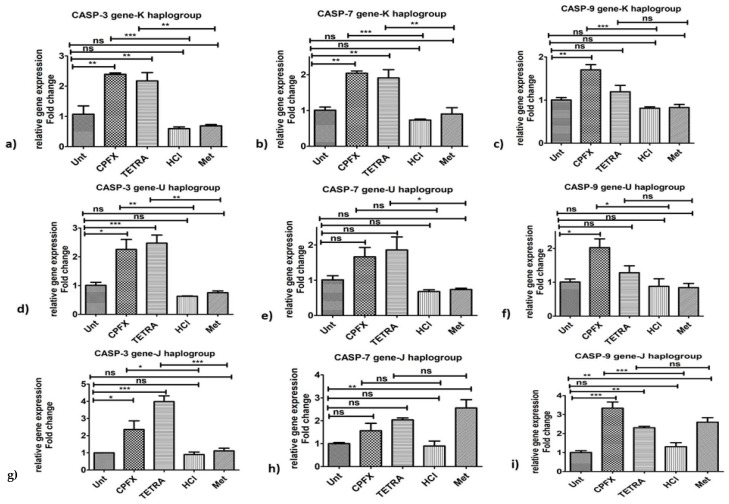
Alterations of pro-apoptotic genes induced by CPFX and TETRA in K, U and J cybrids. Overexpression of *CASP-3* (**a**,**d**,**g**) via CPFX and TETRA in all three cybrids and upregulation of *CASP-7* via CPFX and TETRA in K haplogroup (**b**) and overexpression of *CASP-7* by TETRA in U cybrids (**e**,**h**) and higher *CASP-9* (**c**,**f**,**i**) expression by CPFX in all three cybrids. (* *p* < 0.05, ** *p* < 0.01, *** *p* < 0.0001).

**Figure 5 biomolecules-12-00675-f005:**
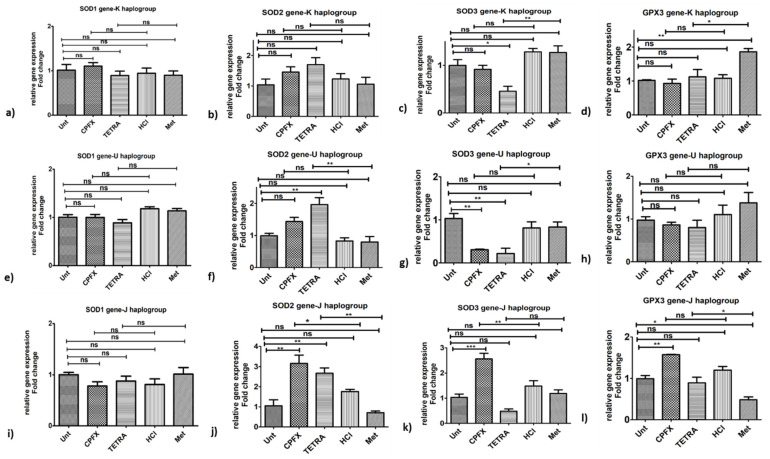
Impacts of CPFX and TETRA on genomic expression of antioxidant enzymes in K, U and J cybrids. In K cybrids, (**a**,**b**) no significant change in *SOD1* and *SOD2* by either treatment, (**c**,**d**) downregulation of *SOD3* and *GPX3* by TETRA. In U cybrids, (**e**,**h**) no significant change in *SOD1* and *GPX3* by either treatment, (**f**) higher expression of *SOD2* and (**g**) lower expression of *SOD3* via TETRA. In J cybrids, (**i**) no significant change in *SOD1* by either treatment, (**j**) upregulation of *SOD2* via CPFX and TETRA, (**k**) higher expression of *SOD3* by CPFX, and (**l**) overexpression of *GPX3* by CPFX. (* *p* < 0.05, ** *p* < 0.01, *** *p* < 0.0001).

**Figure 6 biomolecules-12-00675-f006:**
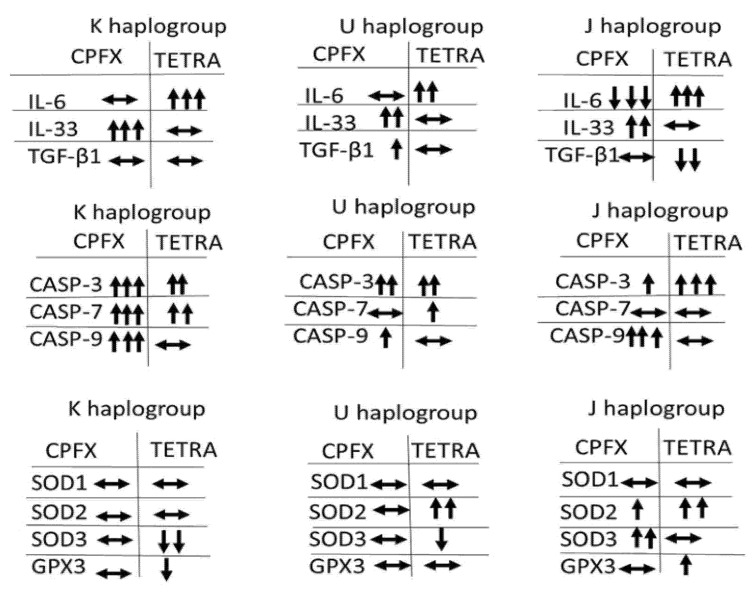
The Summary of Changes in Gene Expression Influenced by CPFX and TETRA Treatment Compared to Vehicle-Control Cells.

**Table 1 biomolecules-12-00675-t001:** Demographics of the K, U and J cybrids.

Cybrid	Haplogroup	Age	Sex	Ethnicity	Diagnosis
16-188	K	90	M	Caucasian	Dry AMD
13-129	K1a1b1a	89	M	Caucasian	Wet AMD
16-187	K2a2a1	82	M	Caucasian	Dry AMD
14-138	U2e1a1	69	M	Caucasian	Dry AMD
17-200	U	69	M	Caucasian	Dry AMD
18-238	U	76	F	Caucasian	Wet AMD
14-136	J2a1a1a2	77	F	Caucasian	Wet AMD
14-142	J1c2g	91	F	Caucasian	Wet AMD

**Table 2 biomolecules-12-00675-t002:** Information of the genes associated with inflammatory, apoptotic and antioxidant enzymes pathways in AMD cybrid cells.

Symbol	Gene Name	GenBank Accession No.	Sigma Primer Sequences Or Qiagen Gene Globe ID	Function
*CASP-3*	Caspase 3, apoptosis-related cysteine peptidase	NM_004346 NM_032991	QT00023947	Encodes protein as a cysteine–aspartic acid protease that plays a central role in the execution phase of cell apoptosis.
*CASP-7*	Caspase 7, apoptosis-related cysteine peptidase	NM_145248, XM_006725153, XM_006725154, XM_005268295, XM_006725155, XM_005268294, XM_006719962	QT00003549	This gene encodes a member of the cysteine–aspartic acid protease (caspase) family. Sequential activation of caspases plays a central role in the execution phase of cell apoptosis.
*CASP-9*	Caspase 9, apoptosis-related cysteine peptidase	NM_001229 NM_032996	QT00036267	Encodes a member of the cysteine–aspartic acid protease (caspase) family, which is involved in the execution phase of cell apoptosis.
*IL-6*	Interleukin 6	NM_000600	FH1-5′-GCAGAAAAAGGCAAAGAATG-3′ RH1-5′-CTACATTTGCCGAAGAGC-3′	This gene encodes a cytokine that functions in inflammation and the maturation of B cells. In addition, the encoded protein has been shown to be an endogenous pyrogen capable of inducing fever in people with autoimmune diseases or infections.
*IL-33*	Interleukin 33	NM_033439 NM_001199640 NM_001127180	FH1-5′-CCAGAAGTCTTTTGTAGG-3′ RH1-5′-GCTGGGAAATAAGGTGTT-3′	The protein encoded by this gene is a cytokine that binds to the IL1RL1/ST2 receptor. The encoded protein is involved in the maturation of Th2 cells and the activation of mast cells, basophils, eosinophils and natural killer cells.
*TGF-β1*	Transforming growth factor beta-1-like	NM_003238	FH1-5′-AACCCACAACGAAATCTATG-3′ RH1-5′-CTTTTAACTTGAGCCTCA-GC-3′	This gene is a polypeptide member of the transforming growth factor beta superfamily of cytokines. It is a secreted protein that performs many cellular functions, including the control of cell growth, cell proliferation, cell differentiation and apoptosis.
*SOD1*	Superoxide dismutase 1	NM_000454	QT01671551	This gene is a member of the iron/manganese superoxide dismutase family. The protein encoded by this gene is a soluble cytoplasmic protein, acting as a homodimer to convert naturally occurring but harmful superoxide radicals to molecular oxygen and hydrogen peroxide.
*SOD2*	Superoxide dismutase 2	NM_000636	FH1-5′-ATCTACCCTAATGATCCCAG-3′ RH1-5′-AGGACCTTATAGGGTTTTCAG-3′	This gene encodes a mitochondrial protein that forms a homotetramer and binds one manganese ion per subunit. This protein binds to the superoxide byproducts of oxidative phosphorylation and converts them to hydrogen peroxide and diatomic oxygen.
*SOD3*	Superoxide dismutase 3	NM-003102	QT01664327	This gene encodes a member of the superoxide dismutase (SOD) protein family, which catalyzes the conversion of superoxide radicals into hydrogen peroxide and oxygen, effective in protection of the brain, lungs and other tissues from oxidative stress.
*GPX3*	Glutathione peroxidase 3	NM_002084	FH1-5′-GCAACCAATTTGGAAAACAG-3′ RH1-5′-CTCAAAGAGCTGGAAATTAGG-3′	The protein encoded by this gene belongs to the glutathione peroxidase family, members of which catalyze the reduction of organic hydroperoxides and hydrogen peroxide (H_2_O_2_) by glutathione, and thereby protect cells against oxidative damage. Several isozymes of this gene family exist in vertebrates, which vary in cellular location and substrate specificity.
*HPRT1*	Hypoxanthine Phosphoribosyl transferase 1	NM_000194	FH1-5′-ATAAGCCAGACTTTGTTGG-3′ RH1-5′-ATAGGACTCCAGATGTTTCC-3′	The protein encoded by this gene is a transferase, which catalyzes conversion of hypoxanthine to inosine monophosphate and guanine to guanosine monophosphate via transfer of the 5-phosphoribosyl group from 5-phosphoribosyl 1-pyrophosphate. This enzyme plays a central role in the generation of purine nucleotides through the purine salvage pathway.

## Data Availability

All data relevant to the study are included in the article.

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
