# Peer review of "Impacts of Bacteriostatic and Bactericidal Antibiotics on the Mitochondria of the Age-Related Macular Degeneration Cybrid Cell Lines"

_biomolecules, 2022, doi:10.3390/biom12050675_

Round 1

Reviewer 1 Report

This paper examines the role of tetracycline and ciprofloxacin on the cybrid cell lines derived from the ARPE-19 cell line.  It is an extension of the authors’ recently published work on Mueller cells and on their earlier research on unmodified ARPE-19 cells.  The authors report adverse impacts on AMD cybrids possessing K, J and U mtDNA haplogroups in cell culture and may indicate that caution should be employed in their clinical application in patients with age related macular degeneration or other diseases affecting mitochondrial function. One serious shortcoming of the paper is that the authors neglected to study the impact of aminoglycoside antibiotics in this study.  Amikacin and gentamicin can cause toxic side effects to the macula, and variations in the coding sequence of the 12S rRNA are associated with ototoxicity. It is also curious that the authors failed to measure expression of SOD1, which is the major isoform of superoxide dismutase and has been implicated in a mouse model of AMD. The statistical analysis in this paper is incorrect: Since the authors are comparing multiple variables, ANOVA with correction for multiple comparisons should have been used, not Student’s t test. Performing the correct test may alter their conclusions.

Other comments:

  1. The authors employ non-standard abbreviations, but the use of MMP for “mitochondrial membrane potential” was the most confusing. In this submission, the authors use the same abbreviation for “matrix metalloproteinase “which is a more common usage. Since both are affected by antibiotics, the authors should use ΔψM to refer to mitochondrial membrane potential, since this is the standard abbreviation.
  2. The tense of the first sentence of the introduction is confusing. The authors should say, “The introduction of antibiotics in the 20th century is considered an astounding breakthrough in medicine."
  3. The authors should mention in the Introduction or in the Discussion that ciprofloxacin inhibits mitochondrial DNA replication and transcription and lowers mtDNA copy number. (PMID: 30169847).
  4. Line 42: Lose the comma after “Although”.
  5. Line 62: Need a new paragraph here, since the topic changes from tetracyclines to mitochondria.
  6. It would be good to begin the Results with a statement reiterating the purpose of the experiment: "To compare the impact of antibiotics on oxidative stress in cells of different mitochondrial haplogroups, we . . .”
  7. All of the figures are all distorted--stretched horizontally-- affecting the axis labeling. This really distorts the flow cytometry results (Fig. 2).

Author Response

Author's Reply to the Review Report (Reviewer 1)

Response# We thank the reviewer for this comment. Our present study is the extended evaluation of two frequently used antibiotics (fluoroquinolones and tetracyclines) in the clinical setting. Since we noticed detrimental impacts of treatments on normal retinal cells including ARPE-19 and Müller cells, we decided to assess these effects on AMD cybrids.

The class of antibiotics such as aminoglycosides (gentamicin and amikacin) pose a significant danger [1]. If aminoglycosides are not administered and monitored properly, they can cause major adverse effects such as permanent hearing loss, vestibular toxicity, and reversible nephrotoxicity. Hearing loss and vestibular dysfunction caused by aminoglycosides are recognized as an issue in underdeveloped nations where the medications are often utilized for their clinical efficacy and affordability [2-3]. Although many clinicians believe that gentamicin is no longer routinely used as a first-line antibiotic in developed nations, ototoxicity remains a significant problem. These antibiotics are rarely used in the clinical settings of develop countries [4]. That is why we exclude this class of antibiotics as it rarely used in the current setting of developed countries.

However, we agree to reviewer point this class of antibiotics has direct effect on the mitochondria. There is scarcity of literature on the effect of aminoglycosides in AMD As a result we are working on a second full-length manuscript to describe these studies on aminoglycosides class of antibiotics and their biochemical and metabolic on AMD.

As per your suggestion, we incorporated the expression level of SOD1 gene, and the results are added to the Figures 5 (a, e, i) of the revised manuscript. Also, we reanalysed the entire data set with one way ANOVA Tukey test. Table number 3 was removed.

Reference: -

  1. New South Wales Government, Ministry of Health. Policy Directive: High- Risk Medicines Management Policy. 2015. https://intranet.nnswlhd.health. nsw.gov.au/docs/PD2015_029-high-risk-medicines-management-policyv-002.
  2. Petersen L, Rogers C. Aminoglycoside-induced hearing deficits – a review of cochlear ototoxicity. South African FamPract 2015; 6190: 1–6.
  3. Murphy JE, Winter ME. The aminoglycoside antibiotics. In: Applied Clinical Pharmacokinetics. 2nd edn.McGraw-Hill, 2008; 97–206.
  4. Chang J, Jung HH, Yang JY, Choi J, Im GJ, Chae SW. Protective role of antidiabetic drug metformin against gentamicin induced apoptosis in auditory cell line. Hear Res. 2011; 282:92–96.

Other comments:

  1. The authors employ non-standard abbreviations, but the use of MMP for “mitochondrial membrane potential” was the most confusing. In this submission, the authors use the same abbreviation for “matrix metalloproteinase “which is a more common usage. Since both are affected by antibiotics, the authors should use ΔψMto refer to mitochondrial membrane potential, since this is the standard abbreviation.

Response 1# We do agree with the reviewer’s comment. Now we have replaced MMP with Δψin the revised manuscript.

  1. The tense of the first sentence of the introduction is confusing. The authors should say, “The introduction of antibiotics in the 20th century is considered an astounding breakthrough in medicine."

Response 2# Thank you for the correction. We modified the sentence. Introduction, Page 2, lines 57-58. It now reads as follows: The introduction of antibiotics in the 20th century is considered an astounding breakthrough in medicine.

  1. The authors should mention in the Introduction or in the Discussion that ciprofloxacin inhibits mitochondrial DNA replication and transcription and lowers mtDNA copy number. (PMID: 30169847).

Response 3# Thank to the reviewer for this comment. We modified the sentence. Introduction, Pages 2 and 3, lines 66-69. It now reads as follows: Their principal mechanism of action is inhibiting DNA topoisomerases and preventing bacterial DNA replication by impending mitochondrial DNA replication, transcription and lowering mtDNA copy number [5,6].

  1. Line 42: Lose the comma after “Although”.

Response 4# We removed the comma. Introduction, Page 3, lines 69-71. It now reads as follows: Although fluroquinolones are well tolerated agents overall, they have induced some serious side effects in some individuals, such as tendinopathy, cardiac arrythmia, and optic neuropathy.

  1. Line 62: Need a new paragraph here, since the topic changes from tetracyclines to mitochondria.

Response 5# As per your suggestion we made as a new paragraph in the revised manuscript.

  1. It would be good to begin the Results with a statement reiterating the purpose of the experiment: "To compare the impact of antibiotics on oxidative stress in cells of different mitochondrial haplogroups, we . . .”

Response 6# As per your suggestion, now we incorporated the results with a “statement reiterating the purpose of experiment” in the revised manuscript.

  1. All of the figures are all distorted--stretched horizontally-- affecting the axis labeling. This really distorts the flow cytometry results (Fig. 2).

Response 7# We do agree with this comment. Figures are re-organized in the revised manuscript.

Reviewer 2 Report

Thanks for providing me an opportunity to review this manuscript. Authors have performed an intersting and a very good study on studying the effect of antibiotics on the mitochondria of the AMD cybrid cell lines. After critical review of the manuscript, my decision is to accept the paper with minor changes.

Authors are advised to perform an editorial check before publishing.

If possible, please provide a graphical abstract for the whole paper. It will be interesting for the readers.

Author Response

Authors are advised to perform an editorial check before publishing.

  1. If possible, please provide a graphical abstract for the whole paper. It will be interesting for the readers.

Response 1# We highly appreciate the reviewer for giving wonderful comments about our study. As per the reviewer suggestion we incorporated the graphical abstract in the revised manuscript.

Reviewer 3 Report

The paper is interesting and well-design, but some changes are required.

  1. I suggest using in the title full name of AMD
  2. All abbreviations, not only in the main text, but also in the abstract have to be explained when they were used at the first time.
  3. The introduction and aim of the study were described clearly and sufficient. 
  4. In the abstract please including all methods used in this study. 
  5. line 106 please add a volume of blood.
  6. Please consider if the section 2.2. should be splitted to 2 sections
  7. Please explain (shortly) why ARPE-19 cell line was chose to this study.
  8. Please write about strengths and limitations of this study in the discussion section.
  9. Please consider to cite DOI: 10.3390/ijerph19042303

Author Response

  1. I suggest using in the title full name of AMD

Response1# We thank the reviewer for this suggestion. The title now has modified to Impacts of bacteriostatic and bactericidal antibiotics on the mitochondria of the age-related macular degeneration cybrid cell lines

  1. All abbreviations, not only in the main text, but also in the abstract have to be explained when they were used at the first time.

Response2# Now we have incorporated the explanation of abbreviations in the revised manuscript

  1. The introduction and aim of the study were described clearly and sufficient.

Response 3# We appreciate the reviewer for the acknowledgment of our study. 

  1. In the abstract please including all methods used in this study.

Response 4# The methods used in this study are mentioned in the abstract. Introduction, Page 2, lines 35-40 “AMD cybrid cells were created and were cultured in 96 well plates and treated with tetracycline (TETRA) and ciprofloxacin (CPFX) for 24 hours. Reactive oxygen species (ROS) levels, mitochondrial membrane potential (ΔψM), cellular metabolism and ratio of apoptotic cells were measured using H2DCFDA, JC1, MTT and flow cytometry assays, respectively. Expression of genes of antioxidant enzymes, pro-inflammatory, and pro-apoptotic pathways were evaluated by Quantitative Real-Time PCR (qRT-PCR).”

  1. line 106 please add a volume of blood.

Response 5# We thank reviewer for raising this point. Now we incorporated the volume of the blood in the revised manuscript. Materials and methods, Page 4, lines 135-138.It now reads as follows: After collecting 10 ml of peripheral blood using tubes with sodium citrate, DNA extraction kits (Puregene, Qiagen, Valencia, CA, USA) and the Nanodrop 1000 (Thermo Scientific, Wilmington, DE, USA) were used for DNA isolation and quantification, respectively.

  1. Please consider if the section 2.2. should be splatted to 2 sections

Response 6# The section 2.2 is splatted. Materials and methods, Page 4, lines 133 and 146. It now reads as follows:  2.2.a Cybrids creation and 2.2.b Cybrids culture conditions

  1. Please explain (shortly) why ARPE-19 cell line was chose to this study.

Response 7 # ARPE19 cells were only used to create rho 0 cells which do not contain mitochondria. These rho 0 cells are fuse with platelets of different patients to develop personalized cybrid. Each cybrid possess the identical nuclei (obtained from rho 0) but different mitochondria.

  1. Please write about strengths and limitations of this study in the discussion section.

Response 8# As per your suggestion now we included the strength and limitation of the study in the discussion section of the revised manuscript.  Strength and Limitations of the study, Page 13, lines 462-466. It now reads as follows:  Strength and limitations of the study: The findings of this study might be significant in clinical settings, particularly regarding approach to treat the patients suffering from specific age-related disease such as AMD, Parkinson disease and Alzheimer’s disease. The only limitation of the study is the smaller cohort of individuals with AMD for three different cybrid cell lines.

  1. Please consider citing DOI: 10.3390/ijerph19042303

Response 9# As per your suggestion we included the reference in the revised manuscript. The reference number is 32.

Round 2

Reviewer 3 Report

The paper has been significantly improved. Good job!